# Laboratory Evaluation of Indigenous and Commercial Entomopathogenic Nematodes against Red Palm Weevil, *Rhynchophorus ferrugineus* (Coleoptera: Curculionidae)

**DOI:** 10.3390/insects15040290

**Published:** 2024-04-19

**Authors:** Mureed Husain, Khawaja G. Rasool, Koko D. Sutanto, Abdalsalam O. Omer, Muhammad Tufail, Abdulrahman S. Aldawood

**Affiliations:** 1Department of Plant Protection, College of Food and Agriculture Sciences, King Saud University, P.O. Box 2460, Riyadh 11451, Saudi Arabia; gkhawaja@ksu.edu.sa (K.G.R.); ksutanto@ksu.edu.sa (K.D.S.); aomer1@ksu.edu.sa (A.O.O.); aldawood@ksu.edu.sa (A.S.A.); 2Department of Plant Protection, Ghazi University, Dera Ghazi Khan 3200, Pakistan; mtufail.ksu@gmail.com

**Keywords:** biological control, RPW, *Steinernema*, *Heterorhabditis*, isolation, pathogenicity bioassay, molecular identification

## Abstract

**Simple Summary:**

The red palm weevil, *Rhynchophorus ferrugineus*, is a major problem for palm plantations worldwide. While chemical pesticides are known to be effective, their environmental impact and insecticide resistance pose serious concerns. Biological control approaches, such as employing entomopathogenic nematodes, can mitigate these challenges by focusing on pests while avoiding harm to the environment or non-target organisms. Our findings show that indigenous and commercial EPNs have the ability to manage red palm weevils. As a result, alternate methods, such as the use of entomopathogenic nematodes, are critical for the sustainable control of red palm weevils.

**Abstract:**

The red palm weevil (RPW) is a significant threat to date palms. Conventional pest control has been ineffective. This study aims to evaluate entomopathogenic nematodes (EPNs) indigenous to Saudi Arabia and commercial against RPW. We used 33 soil samples collected from four areas of Saudi Arabia. The indigenous EPNs were isolated and cultured using an insect baiting method to obtain infective juveniles. Pathogenicity bioassays were conducted against different stages of RPW, including eggs, larvae, and adults. The bioassay was performed using all the isolates at 1 × 10^6^ IJ/mL. Distilled water was used as a control. The results revealed that only 9.09% of soil samples contained positive EPNs. Through DNA sequencing analysis, the positive samples were identified as indigenous isolates belonging to *Heterorhabditis indica* and *Steinernema carpocapsae* EPN species. In pathogenicity tests, 90% mortality of RPW eggs was observed after five days. Similar mortality trends were seen in RPW larvae and adults, with 90% mortality recorded after ten days for all the EPN treatments. Mortality increased with the duration of post-EPN inoculation exposure. The 1 × 10^6^ IJ/mL concentrations of EPN effectively killed various stages of RPW in the laboratory. More research is needed to test EPNs against RPW in the field.

## 1. Introduction

The red palm weevil (RPW) *Rhynchophorus ferrugineus* (Coleoptera: Curculionidae) is a highly detrimental pest that presents substantial peril to palm trees [1,2]. The RPW larvae eat palm tissues, causing damage for several generations before moving to a new tree [3,4]. Infestations can result in significant harm, such as the death of trees [3]. According to reports, the RPW is responsible for causing extensive damage to over ten thousand date palm trees, which are valued in millions of riyals, primarily in the Al Qatif region of Saudi Arabia [5]. At present, chemical insecticides [2,6,7] and pheromone traps [8] are the predominant means to control this crucial insect. However, they do have certain limitations, particularly in terms of insect resistance and environmental issues [9]. The occurrence of insecticide inefficacy is a potential consequence of chemical pesticides, which have caused RPWs to acquire resistance [9,10]. Pesticides, while ineffective against RPW infestations, negatively impact biodiversity and the ecosystem [11]. Biocontrol agents such as entomopathogenic nematodes (EPNs) can be utilized in integrated pest management because ENPs are compatible for use with chemical insecticides [12,13] to control RPW and safeguard the environment [14,15,16].

EPNs can be found all around the world, with each geographic region hosting unique species and groups [17,18]. EPNs are completely safe for both users and the environment [14,19]. They can be easily reared and are capable of quickly detecting, localizing, and infecting a host due to their insecticidal activity [20]. The EPN species, *Steinernema carpocapsae* (Weiser, 1955) (Rhabditida: Steinernematidae) and *Heterorhabditis indica* (Rhabditida: Heterorhabditidae), are frequently utilized for natural pest control [16,19]. The genera *Steinernema* and *Heterorhabditis* have demonstrated the ability to eliminate up to 100% of RPW larvae and adults under laboratory conditions, while in field trials, larval mortality rates of up to 67% have been reported [21]. In laboratory bioassays, *Steinernema* sp. killed 90% of RPW third instar larvae in 72 h, while *Heteroharbitits* sp. achieved 100% larval mortality in field applications [22]. Studies have reported varying levels of mortality in RPW larvae caused by different nematode species [18,23]. 

The primary objective of the present research was to explore and identify local EPNs that could serve as potential alternative biological control strategies against economically significant insect pests, such as the RPW. The second objective was to investigate whether indigenous EPN isolates may exhibit higher pathogenicity towards RPW compared to commercial isolates, as similar findings have been documented [22]. The effectiveness of EPNs in controlling RPW underscores their importance as valuable tools for integrated pest management strategies [4,15]. The present study will explore indigenous EPNs from Saudi Arabia, as well as their efficacy comparison with commercial EPNs against RPW, and provide insights on their potential as biological control agents.

## 2. Materials and Methods

### 2.1. Soil Sample Collection

This study aimed to assess the occurrence and diversity of entomopathogenic nematodes (EPNs) in the Madinah, Tabouk, Al Jouf, and Riyadh regions of Saudi Arabia, where date palm cultivation is more prevalent. A total of 33 soil samples were collected from various vegetation types (see Appendix A). The soil samples were collected from traditional farms that utilized conventional farming practices. Each soil sample weighed one kilogram and was collected from a depth of 0–20 cm [24] using a hand shovel. The samples were placed in plastic bags and stored at 22 °C for further studies.

### 2.2. Isolation of Entomopathogenic Nematodes

From each soil sample, 150 g of soil were taken and placed in a plastic container (15 cm × 9.5 cm × 7.5 cm). The samples were moistened with distilled water to facilitate nematode movement within approximately 80–90% RH. The insect baiting method was employed to extract the EPNs directly from the soil samples [25]. Five larvae of the greater wax moth, *Galleria mellonella*, (Lepidoptera: Pyralidae), and five larvae of the red flour beetle, *Tribolium confusum* (Coleoptera: Tenebrionidae), were introduced into the soil samples in the plastic containers. To prevent moisture loss, the containers were covered and stored at 22 °C. Mortality was monitored every day for 7 days. Appetite loss and larval body colour changes were noted. Figure 1 explains the detailed procedure of nematode infective juvenile (IJ) recovery from the soil samples using the insect baiting method.

### 2.3. Molecular Identification of Entomopathogenic Nematodes

#### 2.3.1. DNA Extraction and Polymerase Chain Reaction Amplification

Nematode DNA was extracted by lysing it with extraction buffer and incubating it at 94 °C for 5 min, as reported by Cimen [26]. The DNA was then separated from the supernatant using alcohol precipitation. Polymerase chain reaction amplification was used to positively identify the nematodes. PCR cycling: initial denaturation at 94 °C for 5 min, then 30 cycles of denaturation at 94 °C for 1 min, annealing at 64 °C for 1 min, and extension at 72 °C for 1 min, ending with a final extension at 72 °C for 5 min. For molecular analysis, primer pairs specific to the genus Steinernema 18S were used to amplify a ribosomal DNA fragment: 5-TTGATTACGTCCCTGCCCTTT-3′ (forward), and 28S: 5-TTTCACTCGCCGTTACTAAGG-3′ (reverse) [27], and *Heterorhabditis* TW81: 5–GTTTCCGTAGGTGAACCTGC–3 (forward), and AB28: 5–ATATGCT TAAGTTCAGCGGGT3 (reverse) [28].

#### 2.3.2. DNA Sequences

For further DNA sequencing analysis, a PCR product of the EPNs was sent to Macrogen Inc. in Seoul, Republic of Korea for sequencing. The obtained sequences were analysed using the Basic Local Alignment Search Tool (BLAST), and the results confirmed the identification of EPNs. Subsequently, the sequences were submitted in the NCBI GenBank database and accession numbers were obtained.

### 2.4. The Red Palm Weevil Rearing for EPN Bioassay

The RPW colony was established in the laboratory at the Plant Protection Department, King Saud University, in a climate chamber (Steridium, Brisbane, Australia), with an ambient temperature of 25 ± 2 °C, a humidity level of 80 ± 5% RH, and a photoperiod of 4 h of light followed by 20 h of darkness (L:D 4:20). The RPW individuals, initially sourced from infested date palms in the Dierab regions, were carefully collected. To ensure optimal conditions, a modified artificial diet was provided to the freshly hatched RPW larvae. The diet composition included date palm frond grinded at 500 g, natrium benzoate at 5 g, ascorbic acid (vit C) at 15 g, sorbic acid at 2 g, corn flour at 250 g, wheat flour at 250 g, and agar at 20 g mixed in 2 L of distilled water [29]. Each larva was then individually housed in cups with dimensions of 5 cm in diameter and 3 cm in height, following the meticulous method described by Al-Ayedh [30]. All stages of RPW, including eggs, larvae, and adults, were methodically collected and utilized for the bioassay pathogenicity test.

### 2.5. Rearing of Entomopathogenic Nematodes

After isolation and identification, the indigenous EPNs were reared using wax moth larvae *Galleria mellonella* Linnaeus (Lepidoptera, Pyralidae) and Tribolium confusum (Duval) (Coleoptera, Tenebrionidae) larvae as the host. The *G. mellonella* and *T. confusum* larvae were obtained from honey bee colonies and the insectarium at the King Saud University, respectively. The last instar larvae of both species were used as a host insect. Five to ten host larvae were placed on filter paper (Whatman No. 1, Macherey-Nagel GmbH & Co. KG, Duren, Germany) in a Petri dish (Sara medical supplies company, Riyadh, Saudi Arabia). The first step was to dilute the EPN juveniles in distilled water. Then, they were pipetted directly onto a wet filter paper in a Petri dish that contained host larvae. To facilitate juvenile movement and breeding, the filter paper was kept moist by daily addition of distilled water. The EPN cultures were maintained in an incubator (Stuart SI500, Tequipment, Long Branch, NJ, USA) at a temperature 25 ± 2 °C and a relative humidity (RH) of 70 ± 5%. However, the commercial EPN isolate acquired from the commercial product “Palmanem” by Koppert (Berkel en Rodenrijs, The Netherlands), which contains the *S. carpocapsae* nematode species, was ready to use for the bioassay immediately after dilution. The concentration of nematodes in both the solutions was measured using a hemocytometer. The nematodes from the commercial product in our storage have shown excellent viability, estimated at around 90–95%. Through careful observation under a microscope during our experiments, we have verified that the nematodes remained active and displayed their expected characteristics.

### 2.6. Pathogenicity Test of EPNs against Red Palm Weevil Eggs, Larvae, and Adults

The pathogenicity test on EPNs was conducted using three different isolates: two indigenous isolates and one commercial isolate obtained from Koppert. Distilled water was used as the control treatment. Red palm weevil eggs (1-day-old) were obtained from the colony for the bioassay. Each treatment of the eggs consisted of five replicates, with each replicate containing five eggs. Five eggs were placed on moistened filter paper in a plastic cup, and 1 mL of EPN solution containing 1 × 10^6^ IJs was poured over the eggs. In the control treatment, 1 mL of distilled water was poured on the eggs. During preliminary trials, several concentrations of commercial nematodes were tried in the lab against RPW stages. The preliminary findings show that a concentration of 106 IJs was effective in killing the RPW stages. As a result, we chose to use this dosage for both commercial and indigenous nematodes as a baseline for our future research.

Similarly, (30-day-old, 8th instar) larvae and (10-day-old) adults were obtained from the colony for the bioassay. There were three replicates for larvae, with each replicate including three larvae and three adults. Each larva was placed singly in a plastic cup to avoid cannibalism and provided with 5 g of the artificial diet. One mL of EPN solution containing 1 × 10^6^ IJs was poured onto the dorsal surface of the larva, following the method described by Abbas [21]. During daily observations after the treatment, it was observed that the nematode juveniles were entering the larval bodies through the cuticle and various openings, such as the mouthparts or anal region.

Similarly, the adults were also individually placed in plastic cups. The adults were provided with a 10% sugar solution as food on a soaked piece of cotton and the EPN IJs were applied to them in the same manner as for the larvae. For the control treatment, the same procedure was followed for all stages, with distilled water used instead of the EPN IJs solution. After the bioassay, all the stages were placed inside the incubator at 25 ± 2 °C, with a humidity level of 80 ± 5% RH, and they were observed at different time intervals.

Daily observations were conducted after the nematodes were applied. To confirm nematode infections, any infected stages were recorded as dead, collected, and placed in sterile Petri dishes lined with moist filter papers. The death of RPW stages due to nematode infections was validated by examining them under a microscope to ensure the emergence of juvenile nematodes from the cadavers. Observations were recorded daily for 5 days for the egg stage and for 15 days for the larvae and adult stages.

### 2.7. Statistical Analysis

Experiments were carried out using a completely randomized design. The recorded mortality percentages of the RPW stages in the bioassay were transformed into angular data transformations [31]. Data were analysed using a one way ANOVA test with the SAS 9.2 program [32], considering each RPW stages’ mortality as a response factor to the nematode infection. Tukey’s test was used to separate the means at *p* < 0.05.

## 3. Results

### 3.1. Isolation and Identification of Entomopathogenic Nematodes

The recovery of EPNs from thirty-three soil samples collected from different areas revealed that EPNs were only found in 9.09% of the samples collected from the Madinah region of Saudi Arabia. On the other hand, soil samples collected from other locations did not show any evidence of EPN presence. This could be due to various factors, such as differences in soil composition, environmental conditions, or agricultural practices that may not be conducive to EPN populations.

The DNA sequences of the positive samples were completely identical to those of the entomopathogenic nematodes and were identified as *Heterorhabditis indica* (Accession number OQ443082) and *Steinernema carpocapsae* (Accession number OQ457611) species.

### 3.2. Entomopathogenic Nematodes Pathogenicity to Red Palm Weevil Eggs

All three examined nematodes were found to be pathogenic to the RPW eggs. *Hetrerorhabditis indica*, an indigenous isolate, exhibited superior performance compared to *S. carpocapsae*, another indigenous isolate, and Palmanem, the commercial product. However, after five days of treatment, there was a notable increase in the mortality rate of the RPW eggs when exposed to all three tested nematodes, showing a significant difference compared to the control group (Table 1).

As time passed following the treatment, the juveniles began to reproduce and multiply within the eggs, establishing a population. After five days, the eggs infected with juveniles were strongly damaged, whereas in the control treatment, embryonic development continued and the eggs hatched into neonates. Figure 2 shows a comparison of nematode infection and embryonic development.

### 3.3. Pathogenicity against Red Palm Weevil Larvae

The results show that all of the indigenous and commercial EPN isolates were highly effective against the RPW larvae. EPN-infective juveniles can enter the larval body by the mouth or anal opening, or they can penetrate through the cuticle and multiply inside. After seven days of inoculation, the commercial *S. carpocapsae* had a maximum larval mortality of 78%. Although not statistically significantly different, it demonstrated better pathogenic efficiency against the RPW larvae compared to the other two isolates. Between seven- and ten-days post-treatment, about 90% of the larvae infected with young EPNs were dead in all treatments except for the control (Table 2). These results demonstrate that both indigenous and commercial EPNs can effectively kill RPW larvae. A comparison between RPW-healthy and EPN-infected larvae is given in Figure 3.

Following the treatment, the juveniles started reproducing and multiplying within the RPW larvae. After seven to ten days, the larvae infected with EPN juveniles were dead and entirely eliminated, while the control larvae fed and grew normally.

### 3.4. Pathogenicity against Red Palm Weevil Adults

All of the studied EPN isolates, whether indigenous or commercial, were effective against the RPW adults. Similar to the larvae, the EPN juveniles were capable of entering the adult body via the mouth or anal opening. After seven days of treatment, the commercial isolate showed superior results, with 90% of adult deaths observed. However, there were no significant statistical differences in cumulative mortality for adults after ten days across all isolates (Table 3). All of the isolates’ results were extremely significant when compared to the control group, which received distilled water as a treatment. A comparison of adult mortality rates for red palm weevils revealed that indigenous EPN isolates could be employed as an effective biological control agent against the RPW.

The EPN infection symptoms examined in the RPW at different developmental stages include obvious evidence of distress, such as increased sluggishness, decreased feeding activity, and abnormal movement patterns. Infected individuals also exhibited physical changes, such as discolouration or darkening of the exoskeleton. These indications suggest that the nematodes had successfully infected and harmed the RPW, resulting in their mortality.

Following treatment, the EPN juveniles began to reproduce and proliferate within the RPW adults. The juvenile build-up was predominantly found on body joints on the ventral side. The most massive juvenile population was found around the base of the cadaver legs and thorax joints, as well as beneath the elytra. Figure 4 depicts the differences between EPN-infected and healthy RPW adults.

## 4. Discussion

The 9% recovery rate of entomopathogens from the collected samples was not only poor but also limited to samples from the Madinah area. Nematode dispersal in various soils requires appropriate soil conditions [33]. Factors such as temperature, moisture, and pH all have an impact on the dispersion of entomopathogenic nematodes [34,35]. EPNs such as *H. indica*, *S. carpocapsae*, and *Acrobeloides saeedi* (Siddiqi, De Ley, and Khan, 1992) (Rhabditida, Cephalobidae) have been found in a few samples from specific locations [34]. The majority of our soil samples were collected from unirrigated farm lands, which may explain the lower presence of EPNs there. However, irrigation and well-maintained fields have a significant impact on EPN survival and dissemination [36]. Many factors influence the presence of EPNs in soil, including soil texture and flora diversity [35]. EPNs such as *H. bacteriophora* are reported to have been present in date palm vegetation [37], which supports our findings.

EPN isolation from soil typically requires sampling and isolation procedures, such as baited traps containing *G. mellonella* [13]. Due to their reliance on insect aggregation, EPNs are more likely to be found in clusters in the soil rather than dispersed uniformly [36]. Our study shows that EPNs were less abundant overall based on the soil samples collected. This finding is similar to those of other studies [38]. However, differences in sampling strategy, insect bait, and soil type can all have a significant impact on the EPN recovery rate [35]. Most research has been conducted in northern provinces like Madinah, Tabouk, Al Jouf, and Riyadh; therefore, the EPNs collected so far may not fully represent Saudi Arabia’s EPNs. In larger samples with more sites, EPNs are more likely to be found, leading to a more diverse population [39]. In 2014, El-Kholy et al. discovered EPNs in Al Jouf, Saudi Arabia [40], which contradicts our findings. Further research is needed to determine why certain entomopathogenic nematodes are present or absent in certain locations.

The recovery frequency of EPNs in the present study was comparable to those reported from other parts of the world. For instance, in Mexico, a recovery frequency of 6.6% (4 positive samples of EPNs out of 60) was reported [41]. Similarly, in China, a recovery frequency of 7% (205 positive samples of EPNs out of 2780) was found [42], while in India, it was 4.9% (5 positive samples of EPNs out of 105) [43]. Evidence of higher recovery frequencies has been reported from Brazil, with 23.2% (73 positive samples of EPNs out of 315) [44], and Mexico, with 29.1% (16 positive samples of EPNs out of 55) [45]. Moreover, soil samples from 102 sites in Egypt have found heterorhabditis nematodes in Alexandria, Behaira, Ismaelia, and Giza with potential for controlling the RPW, but these are less viable in Behaira and Giza [46]. However, expanding surveys to include different geographical areas and climates, such as plantations and forests, is important for discovering new EPN species in European countries. These native EPN species are valuable for biodiversity and the environment as they can adapt to local conditions [47].

Molecular identification is essential for selecting species for biological control programs and addressing taxonomic concerns regarding EPNs [28]. DNA sequence information obtained from the internal transcribed spacer (ITS) gene is commonly used for the molecular characterization of EPNs [37]. The present research showed higher DNA sequence similarity among the three EPN isolates, and this is consistent with the findings of Chaerani., et al. [48].

Moreover, the results of this study show that certain nematode species exhibited rapid infection rates, while all displayed high pathogenicity against the RPW, resulting in significant mortality rates. The high mortality rates observed in the tests suggest that the tested isolates were highly pathogenic to different stages of the RPW. These findings support previous research highlighting the efficacy of EPNs in combating the RPW at various stages [16]. Numerous studies have consistently demonstrated the success of EPNs in controlling both RPW larvae and adults, in both laboratory and field settings [16].

Species of nematodes belonging to the *Steinernema* genus are obligate pathogens that are associated with other pathogens, such as the bacteria *Xenorhabdus nematophila* (Thomas and Poinar (1979)) (Enterobacterales: Morganellaceae) [16]. Our research confirms that these nematode stains have a powerful impact on killing at all stages of the RPW, including eggs, larvae, and adults. This is supported by a study conducted by [49], which further demonstrates the harmful effect of these nematodes on RPW stages. Laboratory conditions have shown that these nematodes can cause up to 100% mortality in RPW larvae and adults [21]. Different isolates of these nematodes have been tested on RPW larvae and adults, and it has been found that *H. bacteriophora* isolate can kill 93–100% of the larvae, while *S. glaseri* (Glaser and Fox, 1930) (Rhabditida: Steinernematidae) and *S. longicaudum* (Shen and Wang, 1992) (Rhabditida: Steinernematidae), *S. carpocapsae* NEMASTAR, and *S. kraussenthei* (Steiner) can kill 100% of RPW larvae; the most effective EPNs were from *S. carpocapsae* [50]. Another study found that *S. glaseri* and *H. bacteriophora* are more effective against RPW larvae, while *S. scapterisci* (Nguyen and Smart, 1990) (Rhabditida: Steinernematidae), *Steinernema* sp., and *S. abbasi* Elawad (Ahmad and Reid, 1997) (Rhabditida: Steinernematidae) are more virulent against adult stages [50]. The study revealed that both imported and locally sourced entomopathogenic nematodes, as well as insecticides, are effective in combating red palm weevil larvae. Among these, Egyptian *H. bacteriophora* have proved to be the most effective, while adult RPWs showed lower susceptibility [51].

Other research studies have shown that *Steinernema* species, such as *S. carpocapsae*, can cause 100% mortality in the RPW 2nd instar larvae [52]. The effectiveness of commercial and indigenous EPNs in controlling the RPW has been evaluated, with mortality rates of 92.20% and 82.13% observed in RPW larvae, particularly with the Egyptian *H. bacteriophora* [50]. Similarly, *H. bacteriophora* has shown remarkable potential in combating the RPW, with larval and pre-adult mortality rates of 80% and 70%, respectively [53]. The combination of *Heterorhabditis* sp. and the entomopathogenic fungus *M. anisopliae* (Metchnikoff) Sorokin (Hypocreales: Clavicipitaceae) has also been found to cause high mortality rates of 92–100% in *Oryctes rhinocheros* (Linnaeus, 1758) (Coleoptera: Scarabaeidae) larvae [54]. Similarly, the combined application of nematodes and insecticides has resulted in high mortality rates in RPW larvae [54]. Additionally, a combination of botanical oils and nematodes has been found to decrease the hatchability of RPW eggs and cause high mortality rates in larvae [54].

Several studies have demonstrated that EPN isolates have varying impacts on different developmental stages. In comparison to *S. feltiae* and *H. bacteriophora*, *S. carpocapsae* is more effective at killing RPW larvae and adults [55]. Both *S. carpocapsae* and *H. bacteriophora* are highly effective nematodes against RPW. They can kill larvae, pupae, and adults with over 90% mortality rates. EPNs can also rapidly eliminate RPW larvae. *S. feltiae* is a potent control method for other coleopteran pests, causing high mortality rates [55]. In contrast, laboratory tests have found that *S. carpocapsae* is more effective in controlling the red palm weevil than *H. bacteriophora* [56].

## 5. Conclusions

The discovery of two indigenous EPN isolates in the Madinah region, including *Heterorhabditis indica* and *Steinernema carpocapsae*, marks the first record from Saudi Arabia. The present findings indicate that both native and commercial EPNs were effective against RPW. The research outcomes show that EPNs could serve as an effective biological control agent for managing RPW, and they can be used as an alternative approach to combating the RPW, a destructive pest of date palm trees worldwide. However, in depth studies are required to evaluate the long-term impact EPNs on RPW populations in the field. These findings, however, are crucial for determining the potential use of nematodes as an alternative to conventional pesticides in reducing red palm weevil infestations. The findings of the present study contribute to current efforts to establish sustainable and effective management techniques for red palm weevil infestations.

## Figures and Tables

**Figure 1 insects-15-00290-f001:**
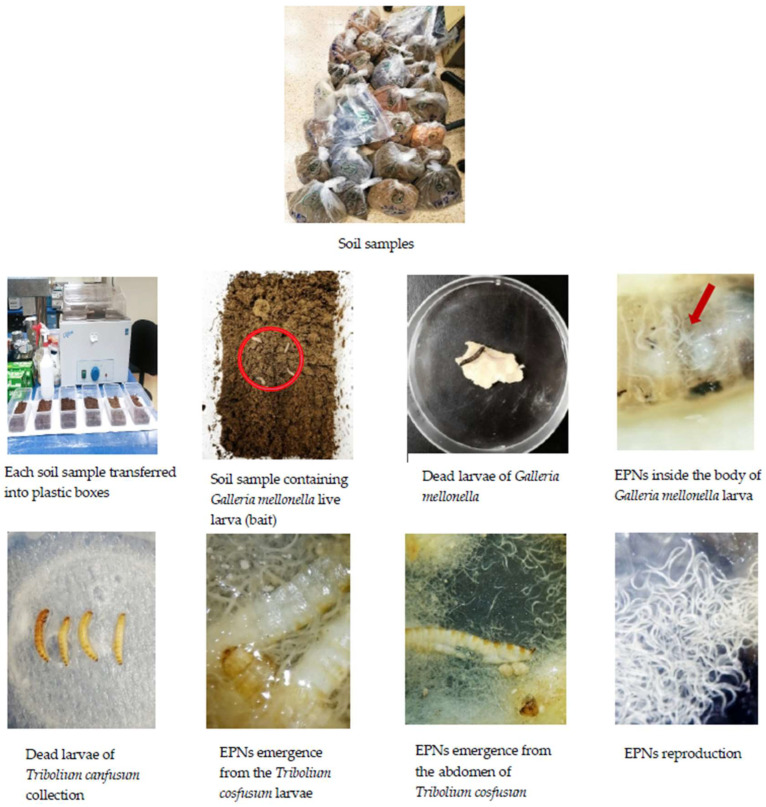
Detailed information of entomopathogenic nematodes isolation/recovery from the soil samples.

**Figure 2 insects-15-00290-f002:**
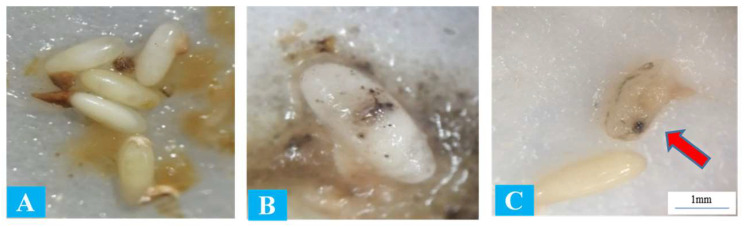
Comparison of red palm weevil eggs: (**A**) healthy eggs exposed to distilled water (control), (**B**) eggs infected by entomopathogenic nematodes, and (**C**) eggs damaged by nematodes. The original egg size was 3.2–3.6 mm in length. The red arrow depicts the infection spot and nematode multiplication.

**Figure 3 insects-15-00290-f003:**
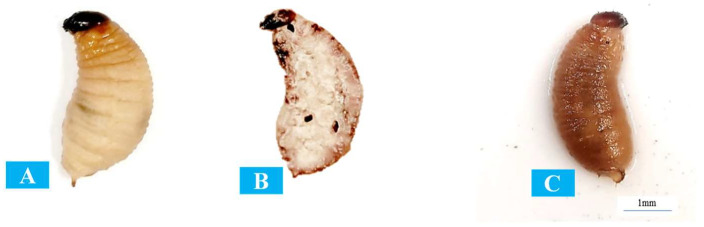
Comparison of red palm weevil larvae: (**A**): healthy larvae exposed to distilled water (control), (**B**) larvae infected with *Steinernema carpocapsae*, and (**C**) larvae infected with *Heterorhabditis indica*. The actual larvae size was measured as 30–35 mm in length.

**Figure 4 insects-15-00290-f004:**
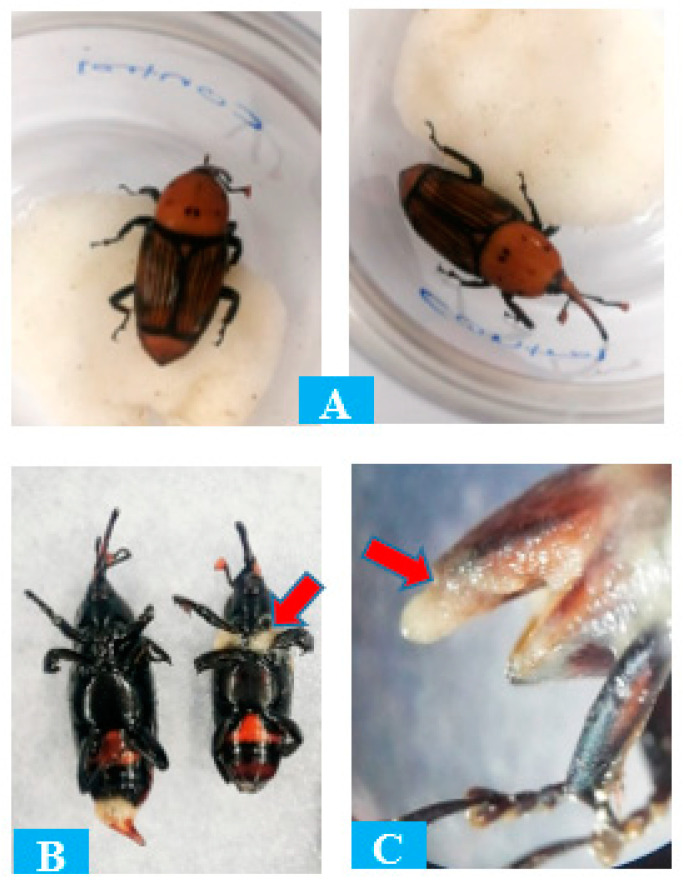
Comparison of healthy and nematode-treated red palm weevil adults: (**A**) healthy adult, exposed to distilled water (control), (**B**) adult infected with nematodes, and (**C**) nematode coming out of the infected adult abdomen. The red arrow depicts the infection spot and nematode multiplication.

**Table 1 insects-15-00290-t001:** Mortality % (mean ± SE) of red palm weevil eggs caused by entomopathogenic nematodes after different exposure times under laboratory conditions.

Exposure Time (Days)	Treatments	Egg Mortality (%)	Statistical Analysis
3	*H. indica*	55.3 ± 8 a	F = 17.7; df = 3, 8; *p* = 0.0007
	*S. carpocapsae*	38.8 ± 6.98 a	
	Palmanem	43 ± 3.86 a	
	Control (H_2_O)	0 ± 0 b	
5	*H. indica*	90 ± 0 a	F = 215, df = 3, 8; *p* ≤ 0.0001
	*S. carpocapsae*	90 ± 0 a	
	Palmanem	90 ± 0 a	
	Control (H_2_O)	1.9 ± 1.9 b	

Means with the same letters do not show significant statistical differences (*p* < 0.05).

**Table 2 insects-15-00290-t002:** Mortality % (mean ± SE) of red palm weevil larvae caused by entomopathogenic nematodes after different exposure times under laboratory conditions.

Exposure Time (Days)	Treatments	Larval Mortality (%)	Statistical Analysis
7	*H. indica*	66.2 ± 11.9 a	F = 14.56, df = 3, 8; *p* ≤ 0.0013
	*S. carpocapsae*	47.9 ± 6.4 a	
	Palmanem	78.1 ± 11.9 a	
	Control (H_2_O)	0 ± 0 b	
10	*H. indica*	90 ± 0 a	F = 215, df = 3, 8; *p* ≤ 0.0001
	*S. carpocapsae*	90 ± 0 a	
	Palmanem	90 ± 0 a	
	Control (H_2_O)	1.9 ± 1.9 b	

Means with the same letters do not show significant statistical differences (*p* < 0.05).

**Table 3 insects-15-00290-t003:** Mortality % (mean ± SE) of red palm weevil adults caused by entomopathogenic nematodes after different exposure times under laboratory conditions.

Exposure Time (Days)	Treatments	Adult Mortality (%)	Statistical Analysis
7	*H. indica*	59.4 ± 16.1 ab	F = 18.50, df = 3, 8; *p* ≤ 0.0034
	*S. carpocapsae*	47.9 ± 6.4 b	
	Palmanem	90 ± 0 a	
	Control (H_2_O)	0 ± 0 c	
10	*H. indica*	90 ± 0 a	F = 215, df = 3, 8; *p* ≤ 0.0001
	*S. carpocapsae*	90 ± 0 a	
	Palmanem	90 ± 0 a	
	Control (H_2_O)	1.9 ± 1.9 b	

Means with the same letters do not show significant statistical differences (*p* < 0.05).

## Data Availability

All relevant data are given in the paper.

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
