# Peer review of "Laboratory Evaluation of Indigenous and Commercial Entomopathogenic Nematodes against Red Palm Weevil, *Rhynchophorus ferrugineus* (Coleoptera: Curculionidae)"

_insects, 2024, doi:10.3390/insects15040290_

Round 1
Reviewer 1 Report
Comments and Suggestions for Authors
Please find comments and suggestions on the attached document.

English needs to be improved.
Author Response
Dear reviewer, thank you so much for your encouragement, support and valuable comments. We really appreciate your effort to review our paper. We revised the manuscript as per your valuable comments and suggestions. Please see the response of your raised comments in the attachment. Kind Regards

Reviewer 2 Report
Comments and Suggestions for Authors
The manuscript titled "Laboratory evaluation of indigenous and exotic entomopathogenic nematodes against the red palm weevil, Rhynchophorus ferrugineus (Coleoptera: Dryophthoridae)" is appropriate for the journal. The research developed by the authors is relevant in the search for sustainable alternatives for the control of Rhynchophorus ferrugineus, through the use of endemic biological control agents. The results are promising to continue with the evaluations.
As listed below, some aspects should be clarified or revised:
Line 93: Table 1: consider as complementary material
Line 98: indicates the approx. humidity level.
Line 106: Figure 1, remove frame
Line 113: describe the PCR amplification conditions
Line 142: No. 1
Line 143: Petri
Line 157: multiplication symbol “×”
Line 163: ml
Line 161: Is the population size sufficient for a bioassay design?
Line 164: multiplication symbol “×”
Line 207: significant statistical differences
Line 214: In Figure 3, insert the letters inside the image
Line 222: no statistically significant difference
Line 243: no statistically significant difference
It is recommended to accept the manuscript after reviewing the comments.

Author Response

(The authors gave the same response as above.)

Reviewer 3 Report
Comments and Suggestions for Authors
This manuscript addresses relevant information about the using of EPNs against a palm pest. My biggest concern is that the authors do not describe sufficiently the soil use. This information it is important to understand the low recovery rate of ENPs at the study site.
Did all growers use the soil on the same way? Were farms organic or conventional?
If this growers using different methods to grow, it is important to compare it and address it in the discussion item..
All photos of insects need a scale and detail which instar is.
More corrections and suggestions in the attached file.

Author Response

(The authors gave the same response as above.)

Round 2
Reviewer 1 Report
Comments and Suggestions for Authors
Please find comments in the attached document.

Minor editing of English is required.
Author Response
Dear reviewer, we would like to express our sincere gratitude to your valuable feedback and insightful comments on our scientific article. Your thorough review has greatly contributed to the improvement of our research, and we truly appreciate the time and effort you dedicated to providing constructive criticism. Thank you for your commitment to advancing scientific knowledge in our field.

Reviewer 3 Report
Comments and Suggestions for Authors
The authors made all corrections or suggestions.
Author Response

(The authors gave the same response as above.)
